# Miniature optical planar camera based on a wide-angle metasurface doublet corrected for monochromatic aberrations

Amir Arbabi[1], Ehsan Arbabi[1], Seyedeh Mahsa Kamali[1], Yu Horie[1], Seunghoon Han[1,2] & Andrei Faraon[1]

Optical metasurfaces are two-dimensional arrays of nano-scatterers that modify optical wavefronts at subwavelength spatial resolution. They are poised to revolutionize optics by enabling complex low-cost systems where multiple metasurfaces are lithographically stacked and integrated with electronics. For imaging applications, metasurface stacks can perform sophisticated image corrections and can be directly integrated with image sensors. Here we demonstrate this concept with a miniature flat camera integrating a monolithic metasurface lens doublet corrected for monochromatic aberrations, and an image sensor. The doublet lens, which acts as a fisheye photographic objective, has a small *f*-number of 0.9, an angle-of-view larger than $60° \times 60°$, and operates at 850 nm wavelength with 70% focusing efficiency. The camera exhibits nearly diffraction-limited image quality, which indicates the potential of this technology in the development of optical systems for microscopy, photography, and computer vision.

[1] T.J. Watson Laboratory of Applied Physics, California Institute of Technology, 1200 E. California Boulevard, Pasadena, California 91125, USA. [2] Samsung Advanced Institute of Technology, Samsung Electronics, Samsung-ro 130, Suwon-si, Gyeonggi-do 443-803, South Korea. Correspondence and requests for materials should be addressed to A.F. (email: faraon@caltech.edu).

Optical systems such as cameras, spectrometers and microscopes are conventionally made by assembling discrete bulk optical components like lenses, gratings and filters. The optical components are manufactured separately using cutting, polishing and grinding, and have to be assembled with tight alignment tolerances, a process that is becoming more challenging as the optical systems shrink in size. Furthermore, the continuous progress of mobile, wearable, and portable consumer electronics and medical devices has rapidly increased the demand for high-performance and low-cost miniature optical systems. Optical metasurfaces offer an alternative approach for realization of optical components[1–5]. Recent advances have increased their efficiency and functionalities, thus allowing metasurface diffractive optical components with comparable or superior performance than conventional optical components[6–10]. The main advantage of metasurfaces stems from the capability to make sophisticated planar optical systems composed of lithographically stacked electronic and metasurface layers. The resulting optical system is aligned lithographically, thus eliminating the need for post-fabrication alignments.

The development of the optoelectronic image sensor has been a significant step towards the on-chip integration of cameras[11]; however, the camera lenses are yet to be fully integrated with the image sensor. The freedom in controlling the metasurface phase profiles has enabled the implementation of spherical-aberration-free flat lenses that focus normally incident light to diffraction limited spots[7,12–14]. Such lenses have been used in applications requiring focusing of an optical beam or collimating emission from an optical fibre[15] or a semiconductor laser[10]. However, the metasurface lenses suffer from other monochromatic aberrations (i.e., coma and astigmatism), which reduce their field of view and hinder their adoption in imaging applications where having a large field of view is an essential requirement. A metasurface lens can be corrected for coma if it is patterned on the surface of a sphere[16–18], but direct patterning of nano-structures on curved surfaces is challenging. Although conformal metasurfaces might provide a solution[19], the resulting device would not be flat. As we show here, another approach for correcting monochromatic aberrations of a metasurface lens is through cascading and forming a metasurface doublet lens.

Here we show that a doublet lens formed by cascading two metasurfaces can be corrected over a wide range of incident angles. We also demonstrate an ultra-slim, low f-number camera, composed of two metasurface lenses placed on top of an image sensor. The camera represents an example of the optical systems enabled by the metasurface vertical integration platform.

## Results

### Design and optimization of the metasurface doublet lens.
Figure 1a schematically shows focusing by a spherical-aberration-free metasurface lens. Simulated focal spots for such a lens are shown in Fig. 1b, exhibiting diffraction limited focusing for normal incidence and significant aberrations for incident angles as small as a few degrees. The proposed doublet lens (Fig. 1c) is composed of two metasurfaces behaving as polarization insensitive phase plates that are patterned on two sides of a single transparent substrate. The aberrations of two cascaded phase plates surrounded by vacuum have been studied previously in the context of holographic lenses, and it has been shown that such a combination can realize a fisheye lens with significantly reduced monochromatic aberrations[20]. We used the ray tracing approach to optimize the phase profiles of the two metasurfaces when they are separated by a 1-mm-thick fused silica substrate. Simulation results of the focal plane spot for different incident angles ($\theta$) are presented in Fig. 1d, showing nearly diffraction

limited focusing by the doublet up to almost 30° incident angle. The doublet lens has an input aperture diameter of 800 μm and a focal length of 717 μm corresponding to an f-number of 0.9. In the optimum design, the first metasurface operates as a corrector plate and the second one performs the significant portion of focusing; thus, we refer to them as correcting and focusing metasurfaces, respectively. The metasurfaces are designed for the operation wavelength of 850 nm, and are implemented using the dielectric nano-post metasurface platform shown in Fig. 2a (ref. 7). The metasurfaces are composed of hexagonal arrays of amorphous silicon nano-posts with different diameters that rest on a fused silica substrate and are covered by the SU-8 polymer. The nano-posts behave as truncated waveguides with circular cross sections supporting Fabry–Pérot resonances[7,9,19]. The high refractive index between the nano-posts and their surroundings leads to weak optical coupling among the nano-posts and allows for the implementation of any phase profile with subwavelength resolution by spatially varying the diameters of the nano-posts. Simulated intensity transmission and phase of the transmission coefficient for different nano-post diameters are presented in Fig. 2b, showing that $2\pi$ phase coverage is achieved with an average transmission over 96% (see Methods for details).

**Device fabrication.** We fabricated the metasurfaces on both sides of a fused silica substrate by depositing amorphous silicon and defining the nano-post pattern using e-beam lithography and dry etching (see Methods for the details). First, the correcting metasurfaces were patterned on the top side of the substrate, and then the focusing metasurfaces were aligned and patterned on the substrate's bottom side (as schematically shown in Fig. 2c). To protect the metasurfaces while processing the other side of the substrate, the metasurfaces were cladded by a layer of cured SU-8 polymer. Aperture and field stops were formed by depositing and patterning opaque metal layers on the top and bottom sides of the substrate, respectively, and anti-reflection layers were coated on both sides of the device. Photos of the top and bottom sides of a set of fabricated metasurface doublet lenses are shown in Fig. 2c. Scanning electron microscope images of the nano-posts are shown in Fig. 2d.

**Focal spot and focusing efficiency characterizations.** We characterized the fabricated metasurface doublet by illuminating it with an 850 nm laser beam at different incident angles (as shown in Fig. 3a), and measuring its focal spot and focusing efficiency. For comparison, a spherical-aberration-free singlet metasurface lens with the same aperture diameter and focal length as the doublet lens (phase profile $\phi(\rho) = -(2\pi/\lambda)\sqrt{\rho^2 + f^2}$, $\rho$: radial coordinate, $f = 717$ μm: focal length, $D = 800$ μm: aperture diameter) was also fabricated and characterized. The focal spots of the metasurface doublet and singlet lenses were measured with two different polarizations of incident light and are shown along with the corresponding simulation results in Fig. 3b,c, respectively (see Methods for details). The doublet lens has a nearly diffraction limited focal spot for incident angles up to more than 25° (with the criterion of Strehl ratio of larger than 0.9, see Supplementary Fig. 1) while the singlet exhibits significant aberrations even at incident angles of a few degrees. As Fig. 3b,c shows, simulated and measured spot shapes agree well. For the doublet lens, a small asymmetry in the 0° spot shape and slightly larger aberrations are observed in the measured spots compared with the simulation results, which we attribute to a misalignment (estimated ∼2 μm along both x and y directions) between the top and bottom side patterns (Supplementary Fig. 2).

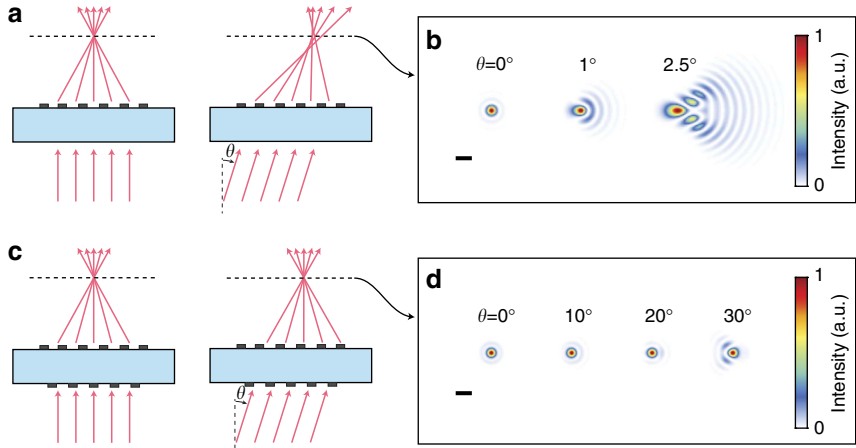

**Figure 1 | Focusing by metasurface singlet and doublet lenses.** (**a**) Schematic illustration of focusing of on-axis and off-axis light by a spherical-aberration-free metasurface singlet lens. (**b**) Simulated focal plane intensity for different incident angles. Scale bar, 2 μm. (**c,d**) Similar illustration and simulation results as presented in **a,b** but for a metasurface doublet lens corrected for monochromatic aberrations. Scale bar, 2 μm. Both lenses have aperture diameter of 800 μm and focal length of 717 μm (*f*-number of 0.9) and the simulation wavelength is 850 nm. See Methods for details.

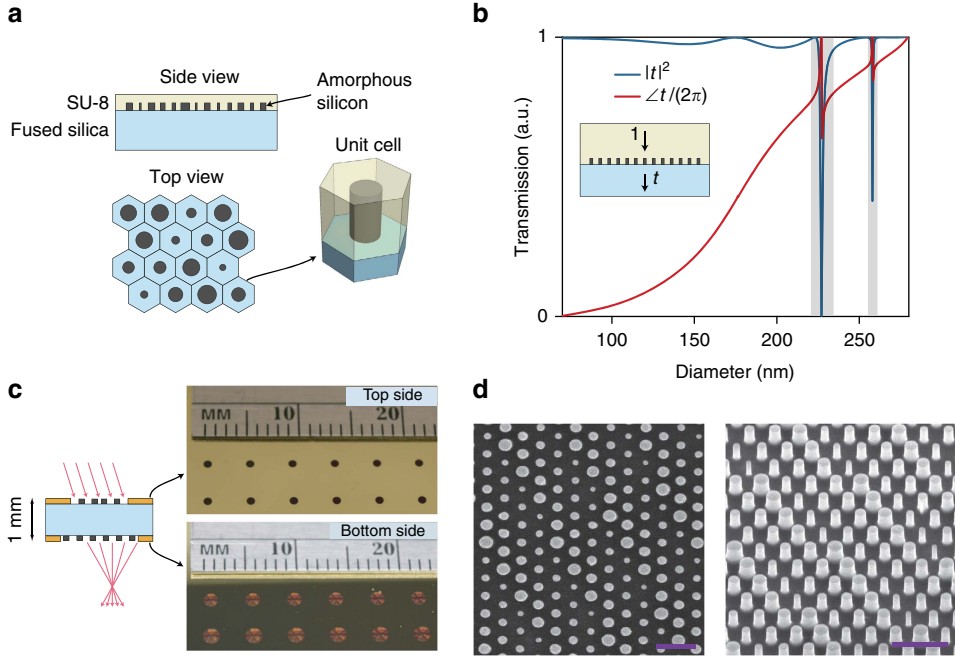

**Figure 2 | Monolithic metasurface doublet lens.** (**a**) A schematic illustration of the dielectric metasurface used to implement the metasurface doublet lens. The metasurface is composed of an array of amorphous silicon nano-posts covered with a layer of SU-8 polymer and arranged in a hexagonal lattice. (**b**) Simulated intensity transmission ($|t|^2$) and the phase of transmission coefficient ($\angle t$) of the metasurface shown in **a** with identical nano-posts as a function of the nano-posts' diameter. The diameters with low transmission values, which are highlighted by two grey rectangles, are excluded from the designs. The nano-posts are 600 nm tall, the lattice constant is 450 nm, and the simulation wavelength is 850 nm. (**c**) Schematic drawing of the monolithic metasurface doublet lens composed of two metasurfaces on two sides of a 1-mm-thick fused silica substrate, an aperture stop and a field stop. The photographs of the top and bottom sides of an array of doublet lenses are also shown. (**d**) Scanning electron micrographs showing a top and an oblique view of the amorphous silicon nano-posts composing the metasurfaces. Scale bars, 1 μm.

The focusing efficiency (ratio of the focused power to the incident power) for the metasurface doublet lens is shown in Fig. 3d, and is ∼70% for normally incident light. The focusing efficiency is polarization dependent, and its value for unpolarized light drops at the rate of ∼1% per degree as the incident angle increases. The measured focusing efficiency at normal incidence is lower than the average of the transmission shown in Fig. 2b because of the large numerical aperture (NA) of the focusing metasurface[7], undesired scattering due to the sidewall roughness of the nano-posts, residual reflection at the air/SU-8 interfaces, and measurement artefacts (see Methods for details). The metasurfaces are polarization insensitive at normal incidence, but their diffraction efficiency depends on the polarization of incident light for non-zero incident angles. The focusing efficiency is lower for the transverse magnetic polarized light compared with the transverse electric polarized light because of the excitation of some resonances of the nano-posts with the axial component of the electric field of the incident light[19]. This also causes the slight difference between the transverse electric and transverse magnetic spot shapes for the 30° incident light shown

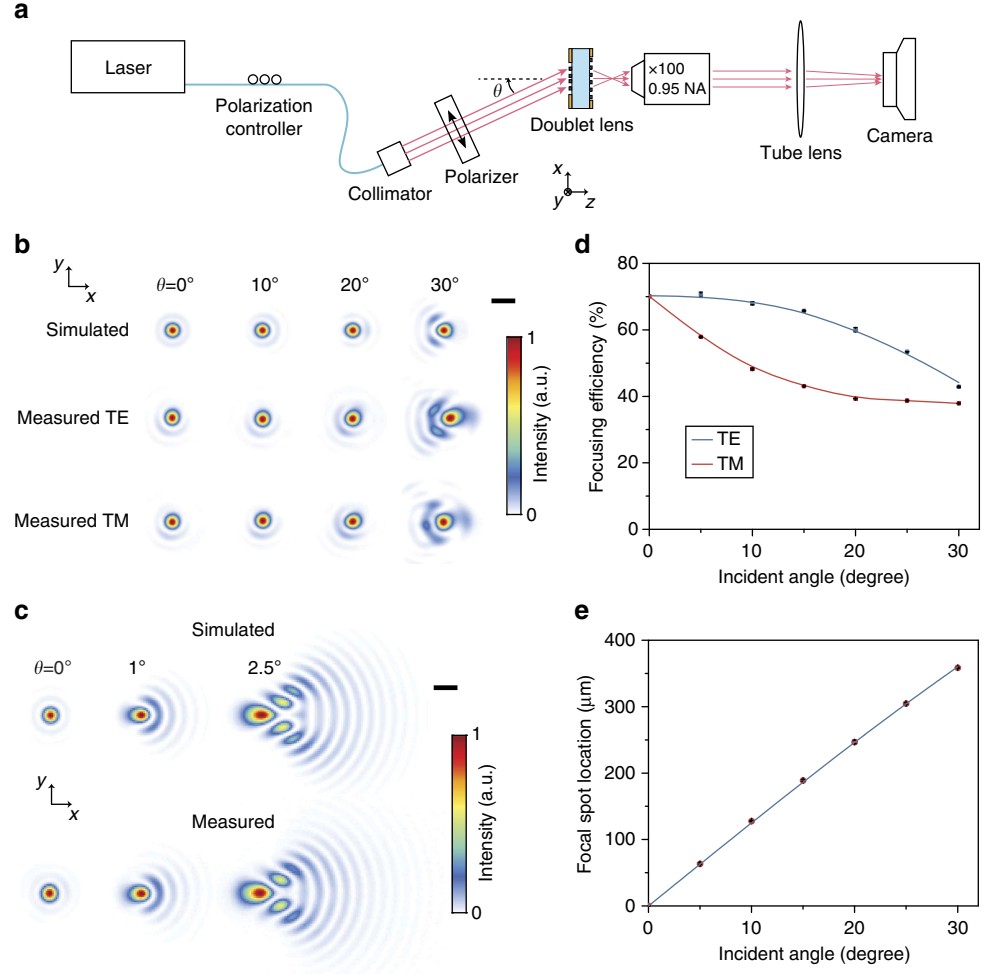

**Figure 3 | Measured and simulated focal spots of the metasurface doublet and singlet lenses.** (**a**) Schematic drawing of the measurement setup. (**b**) Simulated and measured focal plane intensity profiles of the metasurface doublet lens for different incident angles ($\theta$). Simulation results are shown in the top row, and the measurement results for the transverse electric (TE) and the transverse magnetic (TM) polarizations are shown in the second and third rows, respectively. Simulation results are obtained using scalar approximation (that is, ignoring polarization dependence). Scale bar, 2 μm. (**c**) Simulated and measured focal plane intensity profiles for a metasurface singlet with the same aperture diameter and focal length as the metasurface doublet. For the range of angles shown, the measured intensity distributions are polarization insensitive. Scale bar, 2 μm. (**d**) Measured focusing efficiency of the metasurface doublet for TE- and TM-polarized incident light as a function of incident angle. The measured data points are shown by the symbols and the solid lines are eye guides. (**e**) Transverse location of the focal spot for the doublet lens as a function of incident angle. The measured data points are shown by the symbols, and the solid line shows the $f \sin(\theta)$ curve, where $f = 717$ μm is the focal length of the metasurface doublet lens.

in Fig. 3b. We measured a focusing efficiency of $\sim 75\%$ for the singlet, and did not observe a detectable difference between the focal spots measured with transverse electric and transverse magnetic polarizations. The measured relative location of the doublet lens focal spot as a function of incident angle is shown in Fig. 3e along with the $f \sin(\theta)$ curve. The good agreement between the measured data and the curve indicates that the metasurface doublet lens can be used as an orthographic fisheye lens or a wide angle Fourier transform lens[20]. Also, the specific $f \sin(\theta)$ fisheye distortion of the image leads to a uniform brightness over the image plane[21].

**Imaging performance.** We characterized the imaging performance of the metasurface doublet lens using the experimental setup shown in Fig. 4a. A pattern printed on a letter-size paper was used as an object. The object was placed $\sim 25$ cm away from the metasurface doublet lens and was illuminated by an LED (centre wavelength: 850 nm, bandwidth: 40 nm, spectrum shown

in Supplementary Fig. 3). The image formed by the doublet lens was magnified by approximately $\times 10$ using an objective and a tube lens and captured by a camera. A bandpass filter with 10 nm bandwidth (see Supplementary Fig. 3 for the spectrum) was used to spectrally filter the image and reduce the effect of chromatic aberration on the image quality. Figure 4b shows the image captured by the camera, and its insets depict the zoomed-in views of the image at 0°, 15° and 30° view angles. For comparison, an image captured using the same setup but with the metasurface singlet lens is shown in Fig. 4c. The objective lens used for magnifying the images has a smaller NA than the metasurface lenses and limits the resolution of the captured images (see Supplementary Fig. 4 for an image taken with a higher NA objective).

Any imaging system can be considered as low pass spatial filter whose transfer function varies across the field of view. For incoherent imaging systems, the transfer function for each point in the field of view can be obtained by computing the Fourier transform of the focal spot intensity. The modulus of this transfer

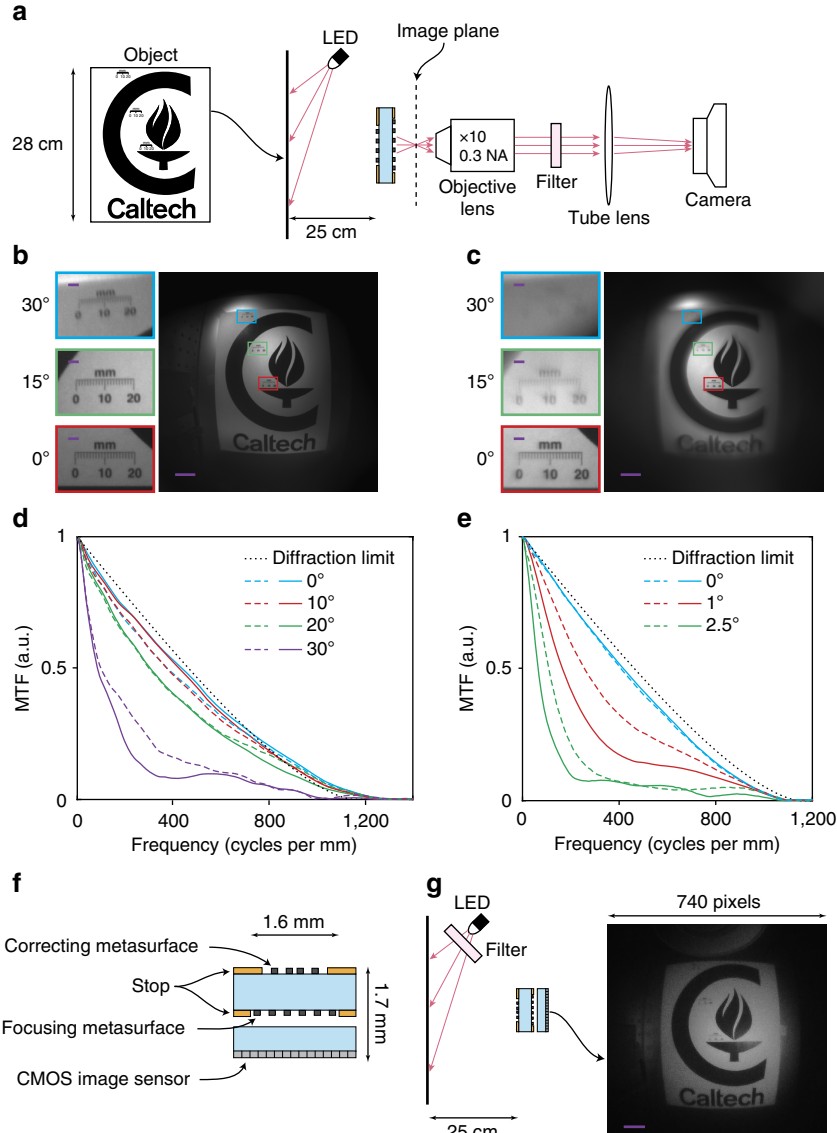

**Figure 4 | Imaging with the metasurface doublet lens.** (**a**) Schematic of the measurement setup. A pattern printed on a letter-size paper is used as the object. The image formed by the metasurface camera is magnified by the combination of the objective lens and the tube lens and is captured by the camera. A bandpass filter (centre wavelength: 850 nm, FWHM bandwidth: 10 nm) is placed between the objective and tube lens to reduce chromatic aberrations. (**b**) Image taken with the metasurface doublet lens and (**c**) with the spherical-aberration-free metasurface singlet lens. Scale bar, 100 μm. The insets show zoomed-in views of the images at the locations indicated by the rectangles with the same outline colours which correspond to viewing angles of 0°, 15° and 30°. Scale bar, 10 μm. (**d**,**e**) Modulation transfer function (MTF) of the metasurface doublet and singlet lenses, respectively. The solid and dashed lines show the MTF in the tangential plane (along x in Fig. 3b) and sagittal plane (along y in Fig. 3b), respectively. The diffraction limited MTF of a lens with aperture diameter of 800 μm and focal length of 717 μm is also shown for comparison. (**f**) Schematic drawing of a miniature planar camera realized using a metasurface doublet lens and a CMOS image sensor. (**g**) Imaging setup and the image captured by the miniature camera. Scale bar, 100 μm. The bandpass filter (centre wavelength: 850 nm, FWHM bandwidth: 10 nm) placed in front of the LED reduces the chromatic aberration.

function is referred to as the modulation transfer function (MTF) and represents the relative contrast of the image versus the spatial details of the object. The MTFs for the metasurface doublet and singlet lenses were computed using the measured focal spots (Fig. 3b,c) and are shown in Fig. 4d,e, respectively. Both the images and the MTFs shown in Fig. 4b–e demonstrate the effectiveness of correction achieved by cascading two metasurfaces, and the diffraction limited performance of the metasurface doublet lens over a wide field of view.

**Miniature metasurface camera**. To further demonstrate the use of this technology in imaging applications, we realized a

miniature planar camera by using a metasurface doublet lens and a CMOS image sensor as schematically shown in Fig. 4f. To compensate for the light propagating through the cover glass protecting the image sensor, another doublet lens was optimized (see Metasurface Doublet Lens II in Supplementary Fig. 5). The total dimensions of the camera (including the image sensor) are 1.6 mm × 1.6 mm × 1.7 mm. The miniature camera was characterized using the setup shown in Fig. 4g and by imaging the object shown in Fig. 4a, which was illuminated by a filtered LED (centre wavelength: 850 nm, bandwidth: 10 nm, see Supplementary Fig. 3 for the spectrum). The image captured by the image sensor is also shown in Fig. 4g, which shows a wide field of view. The camera's image quality is reduced by the

nonuniform sensitivity of the image sensor pixels to the 850 nm light due to the colour filters, and by its larger-than-optimal pixel size. Therefore, the image quality can be improved by using a monochromatic image sensor with a smaller pixel size (the optimum pixel size for the miniature camera is $\sim 0.4\,\mu m$ based on the MTFs shown in Fig. 4d). Thus, the miniature camera benefits from the current technological trend in pixel size reduction.

The intensity of the image formed by a camera only depends on the NA of its lens (it is proportional to $1/f\text{-number}^2 = 4NA^2$ (ref. 22)); therefore, the metasurface miniature camera collects a small optical power but forms a high brightness image. Furthermore, the metasurface doublet lens is telecentric in the image space, and light is incident on the image sensor with the uniform angular distribution (Supplementary Fig. 5), and thus removing the need for the variable incident angle correction in the image sensor.

**Correcting chromatic abberations.** The metasurface doublet lens suffers from chromatic aberrations that reduce the image quality of the miniature camera as the illumination bandwidth increases. Simulated focal spots of the metasurface doublet lens for different illumination bandwidths and the corresponding MTFs are shown in Fig. 5a,b, respectively. See Supplementary Fig. 6 for the off-axis MTFs. As it can be seen from the MTFs, the imaging resolution decreases as the illumination bandwidth increases. This effect can be seen as reduced contrast and lower resolution in the image shown in Fig. 5c (40 nm bandwidth illumination) compared with the image shown in Fig. 5d (10 nm bandwidth illuminations). For the imaging purpose, the fractional bandwidth of a metasurface lens is proportional to $\lambda/(f NA^2)$ (Supplementary Note 1) and can be increased by reducing the NA of the metasurface lens and its focal length. Also, since the MTFs of the metasurface doublet lens shown in Fig. 5b have significant high-frequency components, the unfavourable effect of chromatic aberration can to some extent be corrected using Wiener deconvolution[23]. Figure 5e shows the deconvolution results of the image shown in Fig. 5c that is taken with a 40-nm-bandwidth illumination (see Methods for the details). As expected, the deconvolved image appears sharper and has a higher contrast than the original image; however, deconvolution also amplifies the noise, limiting its applicability for correcting the chromatic aberrations over a significantly wider bandwidth.

**Discussion**

The metasurface doublet lens and camera can be further miniaturized by reducing the thickness of the substrate, the diameters of the metasurface lenses, the focal length of the lens, and the distance to the image sensor by the same scale factor, while using the same nano-post metasurface design presented in Fig. 2. For example, a $10\times$ smaller camera ($160\,\mu m \times 160\,\mu m \times 170\,\mu m$) can be designed and fabricated using a similar procedure on a 100-$\mu m$-thick fused silica substrate. Such a camera would have $10\times$ larger bandwidth compared with the miniature camera presented here, the same image plane intensity, but with $10\times$ smaller image and $100\times$ lower number of distinguishable pixels ($94\times 94$ pixels instead of $940\times 940$). Compared with other miniature lenses reported previously[24–26] and Awaiba NanEye camera (http://www.awaiba.com), the metasurface doublet offers significantly smaller $f$-number and better correction for monochromatic aberrations that lead to brighter images with higher resolution; however, they have larger chromatic aberration (that is, narrower bandwidth).

The miniature metasurface camera concept can be extended for colour and hyperspectral imaging by using a set of metasurfaces that are designed for different centre wavelengths and fabricated side by side on the same chip. Each of the metasurface doublet lenses forms an image on a portion of a single monochromatic image sensor. High-quality thin-film colour filters with different centre wavelengths can be directly deposited on the correcting metasurface of each doublet lens, and the colour filter efficiency issues associated with small size colour filters will be avoided[27,28]. Also, multiwavelength metasurface lenses that work at multiple discrete wavelengths have been demonstrated[29–32]. However, the multiwavelength metasurfaces exhibit the same chromatic dispersion (that is, $df/d\lambda$) and thus similar chromatic aberrations as the single wavelength metasurface lenses. The amorphous silicon metasurfaces have negligible absorption loss for wavelengths above 650 nm. For shorter wavelengths, materials with lower absorption loss such as polycrystalline silicon, gallium phosphide, titanium dioxide[33,34] or silicon nitride[35,36] can be used.

The metasurface-enabled camera we reported here has a flat and thin form factor, small $f$-number, and exhibits nearly diffraction limited performance over a large field of view. From a manufacturing standpoint, the metasurface doublets have several advantages over conventional lens modules. Conventional lens modules are composed of multiple lenses that are separately manufactured and later aligned and assembled together to form the module. On the other hand, the metasurface doublets are batch manufactured with simultaneous fabrication of tens of thousands of doublets on each wafer, and with the metasurfaces aligned to each other using lithographic steps during fabrication. Furthermore, the assembly of the conventional lens modules with the image sensors has to be done in a back-end step, but the metasurface doublet can be monolithically stacked on top of image sensors. More generally, this work demonstrates a vertical on-chip integration architecture for designing and manufacturing optical systems, which is enabled through high performance metasurfaces. This architecture will enable low-cost realization of conventional optical systems (for example, spectrometers, 3D scanners, projectors, microscopes and so on), and systems with novel functionalities in a thin and planar form factor with immediate applications in medical imaging and diagnostics, surveillance and consumer electronics.

**Methods**

**Simulation and design.** The phase profiles of the two metasurfaces composing the doublet lenses were obtained through the ray tracing technique using a commercial optical design software (Zemax OpticStudio, Zemax LLC). The phase profiles were defined as even order polynomials of the radial coordinate $\rho$ as

$$\phi(\rho) = \sum_{n=1}^{5} a_n \left(\frac{\rho}{R}\right)^{2n},\qquad (1)$$

where $R$ is the radius of the metasurface, and the coefficients $a_n$ were optimized for minimizing the focal spot size (root mean square spot size) at incident angles up to 30°. Two different metasurface doublet lenses were designed. The first doublet lens (metasurface doublet lens I) is optimized for focusing incident light in air, and was used in measurements shown in Figs 3 and 4b,d. The second doublet lens (metasurface doublet lens II) is optimized for focusing through the $\sim 445$-$\mu m$-thick cover glass of the CMOS image sensor, and was used in the implementation of the miniature camera as shown in Fig. 4f,g. The optimal values of the coefficients for the two doublet lenses are listed in Supplementary Tables 1 and 2, and the corresponding phase profiles are plotted in Supplementary Fig. 7. The phase profile for the spherical-aberration-free metasurface singlet is given by $\phi(\rho) = -(2\pi/\lambda)\sqrt{\rho^2 + f^2}$, where $f = 717\,\mu m$ is the focal length of the singlet (which is the same as the focal length of the doublet lens I).

The simulation results shown in Figs 1b,d and 2b,c and Supplementary Fig. 2 were computed assuming the metasurfaces operate as ideal phase masks (that is, their phase profile is independent of the incident angle). Incident light was modelled as a plane wave and optical waves passing through the metasurfaces were propagated through the homogeneous regions (that is, fused silica and air) using the plane wave expansion technique[37]. The simulated focal plane intensity results

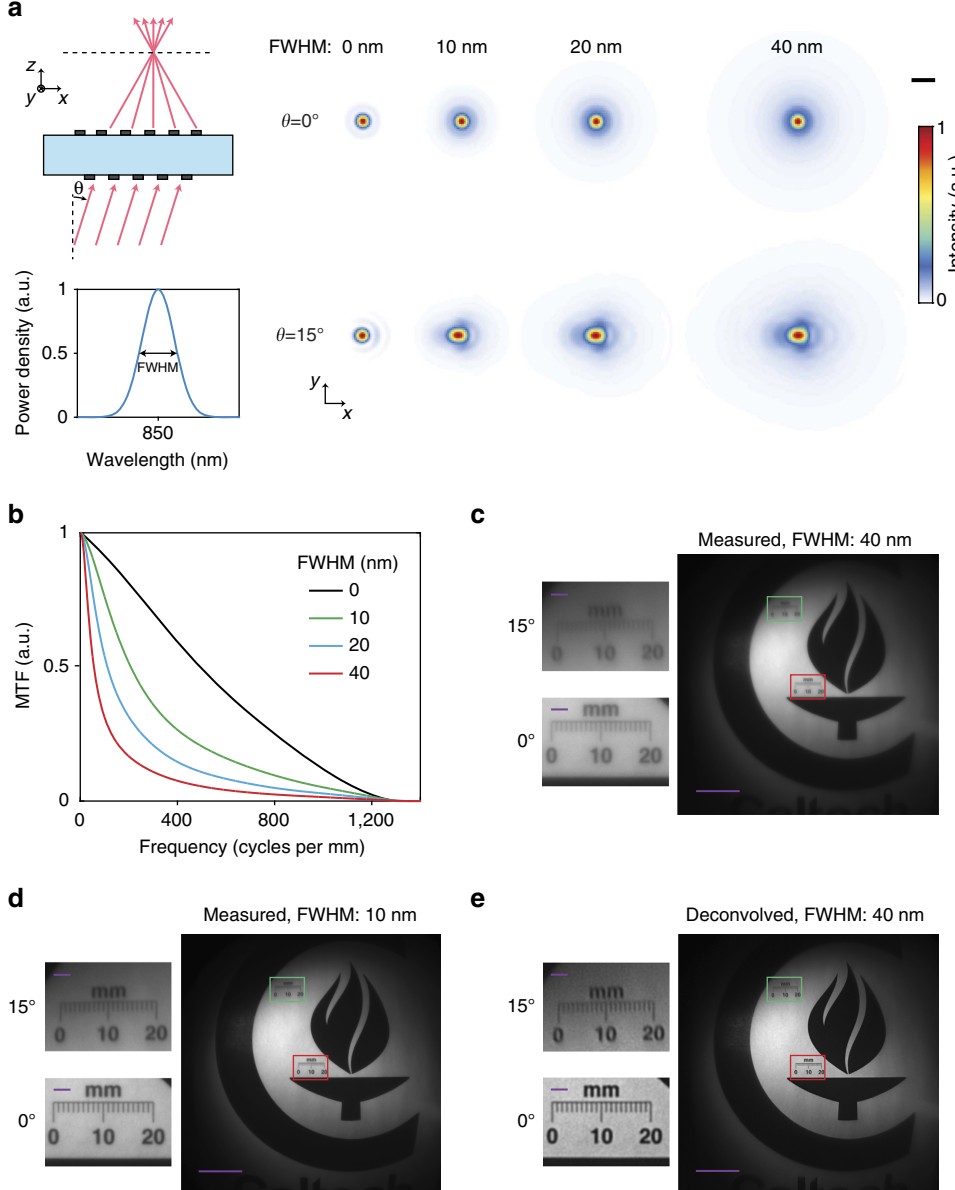

**Figure 5 | Chromatic aberration of metasurface doublet lens and its correction.** (**a**) Illustration of focusing of wideband light by a metasurface doublet lens. The spectrum of the incident light (shown at the bottom) is assumed as a Gaussian function centred at the design wavelength of the doublet lens (850 nm). Simulated focal plane intensity for the metasurface doublet lens as a function of incident light's bandwidth for incident angles of 0° and 15° are shown on the right. Scale bar, 2 μm. (**b**) On-axis modulation transfer function (MTF) of the metasurface doublet lens for different bandwidth of incident light. The MTF is computed using the simulation results shown in **a**. (**c**) Image formed by the metasurface doublet lens with unfiltered LED illumination (40 nm FWHM bandwidth) and (**d**) with filtered LED illumination (10 nm FWHM bandwidth). Scale bar, 100 μm. See Supplementary Fig. 3 for the spectra. The images are captured using the setup shown in Fig. 4a but with a higher magnification objective ( × 20, 0.4 NA). (**e**) Result of chromatic aberration correction for the image shown in **d**. Scale bar, 100 μm. The insets in **c–e** show zoomed-in views of the images at the locations indicated by the rectangles with the same outline colours corresponding to viewing angles of 0° (red border) and 15° (green border), and the scale bars shown in the insets represent 10 μm. FWHM, full width at half maximum.

for wideband incident light, which are shown Fig. 5a, were obtained by computing a weighted average of the optical intensity at several discrete wavelengths in the bandwidth of the incident light. The weights were chosen according to the power density of incident light (Fig. 5a, bottom) assuming the diffraction efficiency of the metasurfaces is constant over the incident light's bandwidth. This assumption is justified because the efficiency of dielectric nano-post metasurfaces does not vary significantly over ∼10% fractional bandwidth[7]. The simulated bandwidth-dependent modulation transfer function of the metasurface doublet lens shown in Fig. 5b and Supplementary Fig. 6 were obtained by taking the Fourier transform of the simulated focal plane intensity distributions presented in Fig. 5a.

Simulated transmission data of the periodic array of amorphous silicon nano-posts presented in Fig. 2b were obtained by using the rigorous coupled wave analysis technique using a freely available software package[38]. The simulations were

performed at $\lambda = 850$ nm. The amorphous silicon nano-posts (with refractive index of 3.6 at 850 nm) are 600 nm tall, rest on a fused silica substrate (refractive index of 1.45), and are cladded with a layer of the SU-8 polymer (refractive index of 1.58 at 850 nm). The imaginary part of the refractive index of amorphous silicon is smaller than $10^{-4}$ at 850 nm and was ignored in the simulations. The nano-posts are arranged in a hexagonal lattice with the lattice constant of $a = 450$ nm. For normal incidence, the array is non-diffractive in both SU-8 and fused silica at wavelengths longer than $\lambda = n_{\text{SU}-8}\sqrt{3}/2a = 616$ nm. The refractive indices of amorphous silicon and SU-8 polymer were obtained through variable angle spectroscopic measurements.

The metasurface patterns were generated using their phase profiles $\phi(\rho)$ and the relation between the transmission and the diameter of the nano-posts shown in Fig. 2b. The diameter of the nano-post at each lattice site ($d$) was chosen to

minimize the transmission error defined as $E = |t(d) - \exp(i\phi)|$, which is the difference between the nano-post transmission $t(d)$ and the desired transmission $\exp(i\phi)$. The nano-posts diameters corresponding to low transmission, which are highlighted in Fig. 2b, are automatically avoided in this selection process, as the low transmission amplitude results in a large transmission error.

**Device fabrication.** The metasurfaces forming the doublet lenses shown in Fig. 2c were fabricated on both sides of a 1-mm-thick fused silica substrate. The substrate was cleaned using a piranha solution and an oxygen plasma. A 600-nm-thick layer of hydrogenated amorphous silicon was deposited on each side of the substrate using the plasma enhanced chemical vapour deposition technique with a 5% mixture of silane in argon at 200 °C. Next, the nano-post patterns for the correcting metasurfaces were defined on one side of the substrate as follows. First a ∼300-nm-thick positive electron resist (ZEP-520A) was spin coated on the substrate and baked at 180 ° C for 5 min. Then a ∼60-nm-thick layer of a water soluble conductive polymer (aquaSAVE, Mitsubishi Rayon) was spin coated on the resist to function as a charge dissipating layer during electron-beam patterning. The metasurface patterns and alignment marks were written on the resist using electron-beam lithography. The conductive polymer was then dissolved in water and the resist was developed in a resist developer solution (ZED-N50, Zeon Chemicals). A 70-nm-thick layer of aluminium oxide was deposited on the resist and was patterned by lifting-off the resist in a solvent (Remover PG, MicroChem). The patterned aluminium oxide was then used as the hard mask in dry etching of the underlying amorphous silicon layer. The dry etching was performed in a mixture of $SF_6$ and $C_4F_8$ plasmas using an inductively coupled plasma reactive ion etching process. Next, the aluminium oxide mask was dissolved in a 1:1 mixture of ammonium hydroxide and hydrogen peroxide heated to 80 °C. Figure 2d shows scanning electron micrographs of the top and oblique view of the nano-posts at this step of the fabrication process. The metasurfaces were then cladded by the SU-8 polymer (SU-8 2002, MicroChem) that acts as a protective layer for the metasurfaces during the processing of the substrate's backside. A ∼2-μm-thick layer of SU-8 was spin coated on the sample, baked at 90 °C for 5 min, and reflowed at 200 °C for 30 min to achieve a completely planarized surface. The SU-8 polymer was then ultraviolet exposed and cured by baking at 200 °C for another 30 min. The complete planarization of the metasurfaces, and the void-free filling of the gaps between the nano-posts were verified by cleaving a test sample fabricated using a similar procedure and inspecting the cleaved cross section using a scanning electron microscope.

The focusing metasurfaces were patterned on the backside of the substrate using a procedure similar to the one used for patterning the correcting metasurfaces. To align the top and bottom metasurface patterns, a second set of alignment marks was patterned on the backside of the substrate and aligned to the topside alignment marks using optical lithography. The focusing metasurface pattern was subsequently aligned to these alignment marks. The aperture and field stops were then patterned by photo-lithography, deposition of chrome/gold (10 nm/100 nm) layers, and photoresist lift-off. To reduce the reflection at the interface between SU-8 and air, a ∼150-nm-thick layer of hydrogen silsesquioxane (XR-1541 from Dow Corning with refractive index of 1.4 at 850 nm) was spin coated on both sides of the substrate and baked at 180 °C for 5 min.

Systematic fabrication errors due to non-optimal exposure dose in e-beam lithography, or over and under etching will generally increase or decrease the nano-post diameters almost by the same amount. To compensate for such errors, we fabricated a set of devices (as shown in Fig. 2c) with all nano-post diameters biased by the same amount (in steps of 5 nm) from their design values. All the devices in the set showed similar focal spots, but with different focusing efficiencies. The focusing efficiency at normal incidence decreased by ∼2.5% for every 5 nm error in the nano-post diameters.

**Measurement procedure and data analysis.** The measurement results shown in Fig. 3b–e were obtained using the experimental setup schematically shown in Fig. 3a. An 850 nm semiconductor laser (Thorlabs L850P010, measured spectrum shown in Supplementary Fig. 3) was coupled to a single mode fibre. The fibre passed through a manual polarization controller and was connected to a fibre collimation package (Thorlabs F220APC-780, $1/e^2$ beam diameter: ∼2.3 mm). The collimated beam was passed through a linear polarizer (Thorlabs LPNIR050-MP) that sets the polarization of light incident on the doublet. The collimator and the polarizer were mounted on a rotation stage whose rotation axis coincides with the metasurface doublet lens. The focal plane of the doublet lens was imaged using an objective lens, a tube lens (Thorlabs AC254-200-B, focal length: 20 cm) and a camera (Photometrics CoolSNAP K4, pixel size: 7.4 μm). A ×100 objective lens (Olympus UMPlanFl, NA = 0.95) was used in measurements shown in Fig. 3b–d, and a ×50 objective lens (Olympus LMPlanFl N, NA = 0.5) with a larger field of view was used for obtaining the focal spot position data shown in Fig. 3e. A calibration sample with known feature sizes was used to accurately determine the magnification of the objective/tube lens combination for both of the objectives. The dark noise of the camera was subtracted from the measured intensity images shown in Fig. 3b,c.

The focusing efficiency presented in Fig. 3d is defined as the ratio of the optical power focused by the lens to the optical power incident on the lens aperture. The focusing efficiency for the normal incidence (zero incident angle) was measured by placing a 15-μm-diameter pinhole in the focal plane of the doublet lens and measuring the optical power passed through the pinhole and dividing it by the power of the incident optical beam. For this measurement, the $1/e^2$ diameter of the incident beam was reduced to ∼500 μm by using a lens (Thorlabs LB1945-B, focal length: 20 cm) to ensure that more than 99% of the incident optical power passes through the aperture of the doublet lens (800 μm input aperture diameter). The incident and focused optical powers were measured using an optical power meter (Thorlabs PM100D with Thorlabs S122C power sensor). The pinhole was a 15-μm-diameter circular aperture formed by depositing ∼100 nm chrome on a fused silica substrate and had a transmission of ∼94% (that is, 6% of the power was reflected by the two fused silica/air interfaces), therefore the reported focusing efficiency values presented in Fig. 3d underestimate the actual values by a few percent.

The focusing efficiency values for non-zero incident angles were found using the focal spot intensity distributions captured by the camera and the directly measured focusing efficiency for normal incidence. First, the focused optical powers for different incident angles were obtained by integrating the focal plane intensity distributions within a 15-μm-diameter circle centred at the intensity maximum. The intensity distributions were captured by the camera when the doublet was illuminated by a large diameter beam ($1/e^2$ beam diameter of ∼2.3 mm) and the dark noise of the camera was subtracted from the recorded intensities before integration. Next, the focused optical powers for different incident angles were compared with the focused power at normal incidence and corrected for the smaller effective input aperture (that is, a $\cos(\theta)$ factor). The measurement was performed for transverse electric (with electric field parallel to the doublet lens's surface) and transverse magnetic (with magnetic field parallel to the doublet lens's surface) polarizations of the incident beam.

The images presented in Fig. 4b,c were captured using the experimental setup schematically shown in Fig. 4a. A pattern was printed on a letter-size paper (∼22 cm × 28 cm) and used as an object. The object was placed in front of the metasurface lens at a distance of ∼25 cm from it. Three ruler marks were also printed as a part of the pattern at viewing angles of 0°, 15° and 30°. The object was illuminated by an 850 nm LED (Thorlabs LED851L, measured spectrum shown in Supplementary Fig. 3). The images formed by the metasurface lenses were magnified by approximately ×10 using an objective lens (Olympus UMPlanFl, ×10, NA = 0.3) and a tube lens (Thorlabs AC254-200-B, focal length: 20 cm). A bandpass filter (Thorlabs FL850-10, centre wavelength: 850 nm, full width at half maximum (FWHM) bandwidth: 10 nm) was placed between the objective lens and the tube lens. The placement of the filter between the objective and the tube lens did not introduce any discernible aberrations to the optical system. The magnified images were captured using a camera (Photometrics CoolSNAP K4) with pixel size of 7.4 μm. The images shown in Fig. 5c,d and Supplementary Fig. 4 were also captured using the same setup but with different objective lenses (Olympus LMPlanFl, ×20, NA = 0.4 for Fig. 5c,d and Olympus LMPlanFl N, ×50, NA = 0.5 for Supplementary Fig. 4).

The miniature camera schematically shown in Fig. 4f is composed of the metasurface doublet lens II (with parameters listed in Supplementary Table 2) and a low-cost colour CMOS image sensor (OmniVision OV5640, pixel size: 1.4 μm) with a cover glass thickness of 445 ± 20 μm. An air gap of 220 μm was set between the metasurface doublet lens and the image sensor to facilitate the assembly of the camera. The metasurface doublet was mounted on a 3-axis translation stage during the measurements. To set the distance between the image sensor chip and the doublet, a far object was imaged and the distance was adjusted until the image was brought into focus.

The modulation transfer functions shown in Fig. 4d,e were computed by taking the Fourier transform of the measured focal plane intensity distributions shown in Fig. 3b,c, respectively. The dark noise of the camera was first subtracted from the recorded intensity distributions. The diffraction limit curves shown in Fig. 4d,e are the simulated modulation transfer function of a diffraction limited lens (that is, Fourier transform of the diffraction limited Airy disk) with the same focal length ($f = 717$ μm) and aperture diameter ($D = 800$ μm) as the metasurface doublet and singlet lenses used in the measurements.

The image shown in Fig. 5e was obtained using Wiener deconvolution[23], and by filtering the image shown in Fig. 5c by the Wiener filter

$$H(v) = \frac{\text{MTF}(v)}{\text{MTF}^2(v) + N(v)/S(v)}, \qquad (2)$$

where $v$ is the spatial frequency, $\text{MTF}(v)$ is the on-axis modulation transfer function of the metasurface doublet lens for illumination with 40 nm FWHM bandwidth (shown in Fig. 5b), and $N(v)$ and $S(v)$ are the power spectral densities of the noise and the image, respectively. The noise was assumed to be white (that is, constant power spectral density), and $S(v)$ was assumed to be equal to the power spectral density of an image formed with a diffraction limited imaging system with NA = 0.4 (that is, the NA of the objective lens used for magnification in the experimental setup). The signal to noise ratio was found as ∼250 by estimating the camera's noise, and was used to set the constant value for $N(v)$.

**Data availability.** The data that support the finding of this study are available from the corresponding author upon request.

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

## Acknowledgements

This work was supported by Samsung Electronics. A.A., E.A. and Y.H. were also supported by DARPA. A.A. was also supported by National Science Foundation award 1512266. Y.H. was supported by a Japan Student Services Organization (JASSO) fellowship. S.M.K., who was involved with the device fabrication, was supported by the DOE 'Light-Material Interactions in Energy Conversion' Energy Frontier Research Centre funded by the US Department of Energy, Office of Science, Office of Basic Energy Sciences under Award no. DE-SC0001293. The device nanofabrication was performed at the Kavli Nanoscience Institute at Caltech.

## Author contributions

A.A and A.F. conceived the experiment. A.A. designed and optimized the device with suggestions from S.H. A.A., E.A., S.M.K. and Y.H. fabricated the sample. A.A. and E.A. performed the measurements and analysed the data. A.A. and A.F. wrote the manuscript with input from all authors.

## Additional information

**Competing financial interests:** The authors declare no competing financial interests.

