## [Peer Review File · Nature Communications]

Reviewers' comments:

Reviewer #1 (Remarks to the Author):

The manuscript titled 'An optical metasurface planar camera' by Arbabi et al, details theoretical and experimental investigations into the development of a miniature camera, dimensions 1.6 x 1.6 x 1.7 mm, employing a 'doublet metasurface' to perform imaging rather than using a conventional lens. They claim it to be 'nearly diffraction limited' over a 60 x 60 degree field-of-view, which to a large extent they evidence in their experimental results. This work is an extension on previously reported work by the authors (most recently 'Multiwavelength polarization-insensitive lenses based on dielectric metasurfaces with meta-molecules, Optica, 2016), however here the authors report a modified design, utilizing two metasurfaces, optimized for 850nm wavelength light.

Demonstration of a working camera of this sort, albeit with very limited wavelength operability, is novel and the results of interest to researchers involved in experimental optics, imaging, and applied physics among other fields.

The manuscript is clearly written and the results are well presented. The results appear to be valid and the methodology is appropriate.

I have some concerns about some of the claims the authors make regarding the impact of metasurfaces in their intended applications, since they do not make clear what the scale of the cost saving to be compared to more conventional imaging technologies. Or if instead it is the small form factor that is the key strength, how small can these 'metasurface planar cameras' be?

There are a few specific issues the authors should address by making modifications to the manuscript or by clarifying in their response, after which I would consider this work suitable for publication in Nature Communications.

Detailed comments:

- page 3, paragraph 2: the authors state that the metasurfaces are polarization insensitive yet on page 4, paragraph 3 they describe the focusing efficiency as polarization sensitive. This needs better clarification.
- page 4, paragraph 2: based on the results shown in Fig 3b, one could argue that the at 30 degrees the focused spot does not look diffraction limited, but instead closer to 20 degrees, but in comparison to the singlet design there is certainly a significant improvement. Therefore, what criteria did the authors use to determine when exactly the spot is not considered diffraction limited?
- page 4, paragraph 3: to determine the focusing efficiency the authors use the ratio of focused power to incident power, but do not explain if a mask is used to determine the area of interest for this measurement. Please can they clarify this and provide detail as necessary.
- page 5, paragraph 2: the authors indicate that a spectral filter is employed subsequent to the metasurface, but in later investigations whereupon the metasurface is in close proximity to the sensor, the filter is placed after the illumination before the metasurface. Can the authors comment on whether they would expect any variation in their results, specifically regarding the MTF.
- page 5, paragraph 2: the authors should provide another sentence or two when they introduce modulation transfer functions, to clearly outline their subsequent use within the manuscript, or at least provide further details in the supplementary materials.
- page 6, paragraph 1: the authors indicate that a smaller pixel size of 0.4 micrometers would lead to improved image quality based on the MTFs, theorizing that eventually pixels will reach this scale. The authors must be aware that this is an order of magnitude smaller than today's miniature sensors, which are themselves struggling for SNR due to reduced light gathering power compared to the noise floor. I recommend a cautionary sentence to add to this paragraph, which at least acknowledges this.

- In neither the introduction nor discussion to put their work into context, do the authors make any reference to other, more conventional but comparatively smaller cameras, which arguably do not have such limited operational spectral bandwidth (e.g. <https://www.fraunhofer.de/en/press/research-news/2011/march/cameras-out-of-the-salt-shaker.html>). Perhaps they should consider highlighting briefly the 'state of the art' in miniature cameras, or detail the specific limitations that metasurface based devices will potentially overcome.

Reviewer #2 (Remarks to the Author):

Dielectric metasurfaces for the visible spectral range are promising candidates for novel integrated devices and optical elements. Here, Arabi et al. present a similar concept for realizing a metalens as recently published by Khorasaninejad et al. in Science. However, the work distinguishes slightly as it does not use the Berry phase effect for generating the phase pattern but a propagation effect to obtain the desired phase. A further difference is the material system. The authors used silicon nanostructures in their study which are easier to handle in the fabrication process and would also have advantages for a real commercial application by easier integration with silicon based cameras. I found the idea and demonstration of the entire imaging system with the camera chip very impressive. In my opinion this is the first demonstration of a real application with metasurfaces. Such demonstration of integrated devices was not obtained by the Harvard group.

The work nicely demonstrates the potential of such dielectric metasurfaces and the realization of a doublet lens system underlines the power of such approaches. Therefore, I will recommend the publication of the manuscript in Nature Communications. I personally believe that the presented approach here will have a greater impact on real imaging systems for particular applications than the recently published work by Khorasaninejad. Overall, the manuscript is well written and the details for the fabrication process are extensively explained. The discussion of the transfer function gives a good insight into the performance of the device. However, I would recommend to move part of the fabrication and the measurement procedure to the supplementary file.

There are only a few comments for the authors which they should take into account for a revision:

On page 6 it is stated that 'The intensity of the image formed by a camera only depends on the NA of its lens.' This statement is in my opinion wrong. The intensity of the image is given by the ratio of the diameter of the entrance pupil to the focal length, which is the inverse f-number.

The distance between the metalens and the CCD chip seems to be important and it was taken care about that in the design as demonstrated in the manuscript. However, I could not find any statement of how the authors did this precise alignment for the measurement? Here it would be helpful to add some information.

It seems like the simulations were performed only with a real part of the refractive index. What is the influence of the imaginary part here? Why can it be neglected?

Supplementary Figure 2a shows the laser spectrum. Why is there such a strong modulation in the spectrum that looks like an interference effect?

Reviewer #3 (Remarks to the Author):

Authors report in this paper a compact camera that utilizes a flat metasurface doublet lens to deliver nearly diffraction-limited performance within the field-of-view of 60 degree x 60 degree. The overall dimensions of the camera (including the image sensor) are 1.6 mm x 1.6 mm x 1.7 mm. The metasurface doublet lens consists of one metasurface corrector plate and one metasurface focusing lens. The phase profiles of both metasurface lenses have been optimized to collectively reduce the monochromatic aberrations. As the results, the performance of the metasurface doublet has been significantly improved in comparison with the singlet lens. This is a significant step in developing a

high performance flat lens for the purpose of optical imaging, as opposite to the focusing being demonstrated before. Additionally, the metasurface doublet has been conveniently fabricated on the both side of 1mm thick quartz substrate with the alignment accuracy of 2 μm . It eliminates the post-fabrication alignment procedure and thus, makes it possible for potential vertical integration using the well-established micro-fabrication capabilities. Just for the curiosity, can author explain of the whether the functions of the two metasurface lenses can be combined into one metasurface lens with aspherical phase profile? Furthermore, the title of "An optical metasurface planar camera" is not very accurate. The demonstrated camera has the shape close to a cubic so it is hard to define it as a "flat camera". The lens being used is the flat one though. Overall, the manuscript is well written and the reported work is of the great interest to the readers. I would recommend the paper to be accepted for publication with minor revision.

Reviewer #4 (Remarks to the Author):

The authors describe in their paper a planar single-layer and double-layer lens based on optical metasurfaces. Furthermore they combine their proposed metasurface doublet lens with a commercial CMOS image sensor.

The used approach for producing the metasurface lenses is not restricted to laboratory prototypes. A high-volume fabrication with thin-film production line is possible, which might yield to a high economic impact of the proposed lenses.

Additionally the authors combine their proposed lens with a commercial CMOS image sensor. Compared to the state-of-the-art they replace a "classical" lens with their proposed one. However the authors should more clearly explain the advantages of their lens for the camera module. From a production point of view for the camera module I do not see a significant advantage. The assembly of the lens together with the CMOS sensor still needs to be done in the backend with a very similar process.

A topic not mentioned in the paper is the influence of the alignment of the nano-posts of the lens and the bayer pattern of the CMOS sensor.

Some further remarks:

- Fig. 2b: what's the reason for the low transmission values
- Non consistent wording for the substrate of the lens: "fused silica" vs. "glass" might be a bit confusing
- p. 7: "high-throughput nano-fabrication techniques" is in my opinion a bit misleading. The fabrication of the lens together with the CMOS sensor in the frontend is in my opinion not feasible. Instead both have to be produced separately in different frontend processes and then combined in the backend

All in all the paper describes a novel and interesting approach for planar lenses which the possibility for a high impact. In addition the authors describe a planar camera module with their planar lenses. This approach is also interesting, but the advantages of the proposed solution are not completely clear for me.

Our response to the reviewers' comments and the corresponding text from the manuscript are presented below in blue and green fonts, respectively.

Reviewer #1 comment:

The manuscript titled 'An optical metasurface planar camera' by Arbabi et al, details theoretical and experimental investigations into the development of a miniature camera, dimensions 1.6 x 1.6 x 1.7 mm, employing a 'doublet metasurface' to perform imaging rather than using a conventional lens. They claim it to be 'nearly diffraction limited' over a 60 x 60 degree field-of-view, which to a large extent they evidence in their experimental results. This work is an extension on previously reported work by the authors (most recently 'Multiwavelength polarization-insensitive lenses based on dielectric metasurfaces with meta-molecules, Optica, 2016), however here the authors report a modified design, utilizing two metasurfaces, optimized for 850nm wavelength light.

Demonstration of a working camera of this sort, albeit with very limited wavelength operability, is novel and the results of interest to researchers involved in experimental optics, imaging, and applied physics among other fields.

The manuscript is clearly written and the results are well presented. The results appear to be valid and the methodology is appropriate.

Our response:

We thank the reviewer for carefully reading and summarizing the manuscript.

Reviewer #1 comment:

I have some concerns about some of the claims the authors make regarding the impact of metasurfaces in their intended applications, since they do not make clear what the scale of the cost saving to be compared to more conventional imaging technologies.

Our response:

The metasurface camera lenses we present in the manuscript have several advantages over conventional bulk lenses, namely high imaging quality with small and flat form factor, small f-number, high scalability of the fabrication process allowing for batch fabrication of tens of thousands of camera lenses on a same wafer, and elimination of post-fabrication alignment and assembly steps required for fabrication of camera lenses. We are unable to provide an accurate cost saving figures at this stage of the project, but we expect the proposed camera lens to be more cost effective than the conventional solution with the similar specifications because the wafer level fabrication significantly benefits from the economy of scale.

We have now further clarified the advantages of the metasurface doublet over conventional solution:

"The metasurface-enabled camera we reported here has a flat and thin form factor, small f-number, exhibits nearly diffraction limited performance over a large field of view. From a manufacturing standpoint, the metasurface doublets have several advantages over conventional lens modules. Conventional lens modules are composed of multiple lenses which are separately manufactured and later aligned and assembled together to form the module. On the other hand, the metasurface doublets are batch manufactured with simultaneous fabrication of tens of thousands of doublets on each wafer,

and with the metasurfaces aligned to each other using lithographic steps during fabrication. Furthermore, the assembly of the conventional lens modules with the image sensors has to be done in a back-end step, but the metasurface doublet can be monolithically stacked on top of image sensors.”

Reviewer #1 comment:

Or if instead it is the small form factor that is the key strength, how small can these ‘metasurface planar cameras’ be?

To address the reviewer’s question regarding the size scaling of these cameras, we added the following to the Discussion section of the manuscript:

“The metasurface doublet lens and camera can be further miniaturized by reducing the thickness of the substrate, the diameters of the metasurface lenses, the focal length of the lens, and the distance to the image sensor by the same scale factor, while using the same nano-post metasurface design presented in Fig. 2. For example, a 10× smaller camera ($160\ \mu\text{m} \times 160\ \mu\text{m} \times 170\ \mu\text{m}$) can be designed and fabricated using a similar procedure on a 100- μm -thick fused silica substrate. Such a camera would have 10× larger bandwidth compared to the miniature camera presented here, the same image plane intensity, but with 10× smaller image and 100× lower number of distinguishable pixels (94×94 pixels instead of 940×940).”

Reviewer #1 comment:

There are a few specific issues the authors should address by making modifications to the manuscript or by clarifying in their response, after which I would consider this work suitable for publication in Nature Communications.

Detailed comments:

- page 3, paragraph 2: the authors state that the metasurfaces are polarization insensitive yet on page 4, paragraph 3 they describe the focusing efficiency as polarization sensitive. This needs better clarification.

Our response:

The metasurfaces are polarization insensitive upon normal incidence. This is similar to transmission through the planar interface between two materials which is polarization insensitive upon normal incidence, but depends on the polarization for non-zero incident angles. We refer to our metasurface design as polarization insensitive to distinguish them from metasurfaces which work only with one polarization (such as the ones which use geometric phase). For clarification, we added the following to the manuscript:

“The metasurfaces are polarization insensitive at normal incidence, but their diffraction efficiency depends on the polarization of incident light for non-zero incident angles.”

Reviewer #1 comment:

- page 4, paragraph 2: based on the results shown in Fig 3b, one could argue that the at 30 degrees the focused spot does not look diffraction limited, but instead closer to 20 degrees, but in comparison to the singlet design there is certainly a significant improvement. Therefore, what criteria did the authors use to determine when exactly the spot is not considered diffraction limited?

Our response:

We thank the reviewer for bringing up this point and we agree with them that a quantitative measure should be used for determining nearly diffraction limited focusing. A widely accepted metric for the focusing quality is Strehl ratio which is the ratio between the volume under the 2D MTF of a lens to the volume under the 2D MTF of the diffraction limited lens with the same NA. We computed the Strehl ratio for the doublet and singlet and added them as Supplementary Fig.1. We chose a threshold of 0.9 for the Strehl ratio to refer to a focal spot as nearly diffraction limited. With this criterion, as the Supplementary Fig. 1 shows, the designed doublet is nearly diffraction limited up to more than 25° (but less than 30°). In addition to adding the Supplementary Fig. 1, we revised the manuscript and added the following explanation:

“The doublet lens has a nearly diffraction limited focal spot for incident angles up to more than 25° (with the criterion of Strehl ratio of larger than 0.9, see Supplementary Fig. 1) while the singlet exhibits significant aberrations even at incident angles of a few degrees.”

Reviewer #1 comment:

- page 4, paragraph 3: to determine the focusing efficiency the authors use the ratio of focused power to incident power, but do not explain if a mask is used to determine the area of interest for this measurement. Please can they clarify this and provide detail as necessary.

Our response:

We used a mask when measuring the focusing efficiency and the procedure was detailed in the Methods section (second and third paragraphs of the “Measurement procedure and data analysis” subsection):

“The focusing efficiency for the normal incidence (zero incident angle) was measured by placing a 15- μm -diameter pinhole in the focal plane of the doublet lens and measuring the optical power passed through the pinhole and dividing it by the power of the incident optical beam.”

Reviewer #1 comment:

- page 5, paragraph 2: the authors indicate that a spectral filter is employed subsequent to the metasurface, but in later investigations whereupon the metasurface is in close proximity to the sensor, the filter is placed after the illumination before the metasurface. Can the authors comment on whether they would expect any variation in their results, specifically regarding the MTF.

Our response:

Placing the filter between the objective and the tube lens did not create any detectable aberrations because the objective lenses used in the measurements were infinity corrected. We added the following to the Methods section to clarify this:

“The placement of the filter between the objective and the tube lens did not introduce any discernible aberrations to the optical system.”

Reviewer #1 comment:

- page 5, paragraph 2: the authors should provide another sentence or two when they introduce modulation transfer functions, to clearly outline their subsequent use within the manuscript, or at least provide further details in the supplementary materials.

Our response:

Per reviewer’s request, we expanded the explanation of the MTF by adding the following to the manuscript:

“Any imaging system can be considered as low pass spatial filter whose transfer function varies across the field of view. For incoherent imaging systems, the transfer function for each point in the field of view can be obtained by computing the Fourier transform of the focal spot intensity. The modulus of this transfer function is referred to as the modulation transfer function (MTF) and represents the relative contrast of the image versus the spatial details of the object.”

Reviewer #1 comment:

- page 6, paragraph 1: the authors indicate that a smaller pixel size of 0.4 micrometers would lead to improved image quality based on the MTFs, theorizing that eventually pixels will reach this scale. The authors must be aware that this is an order of magnitude smaller than today’s miniature sensors, which are themselves struggling for SNR due to reduced light gathering power compared to the noise floor. I recommend a cautionary sentence to add to this paragraph, which at least acknowledges this.

Our response:

In contrast to the reviewer’s comment, we do not theorize or predict that eventually pixels will reach 0.4 μm . The part of the manuscript the reviewer is referring to read as:

“The camera's image quality is reduced by the nonuniform sensitivity of the image sensor pixels to the 850 nm light due to the color filters, and by its larger-than-optimal pixel size. Therefore, the image quality can be improved by using a monochromatic image sensor with a smaller pixel size (which is 0.4 μm based on the MTFs shown in Fig. 4d). Thus, the miniature camera benefits from the current technological trend in pixel size reduction.”

In the above statement we claim that:

1. The optimum pixel size for the miniature camera is 0.4 μm
2. The miniature camera benefits from reducing pixel size
3. There is a technological trend in pixel size reduction

We agree with the reviewer that there are technological challenges in pixel size reduction, however; we note that the pixel sizes of today's miniature sensors are not an order of magnitude larger than 0.4 μm . For example, the CMOS image sensor we used in our study has pixel size of 1.4 μm , or the pixel size for the Samsung's S5K3L2 image sensor which is used in current cell phone cameras is equal to 1.12 μm .

We revised the statement to eliminate any potential ambiguity:

"Therefore, the image quality can be improved by using a monochromatic image sensor with a smaller pixel size (the optimum pixel size for the miniature camera is 0.4 μm based on the MTFs shown in Fig. 4d). Thus, the miniature camera benefits from the current technological trend in pixel size reduction."

Reviewer #1 comment:

- In neither the introduction nor discussion to put their work into context, do the authors make any reference to other, more conventional but comparatively smaller cameras, which arguably do not have such limited operational spectral bandwidth (e.g. <https://www.fraunhofer.de/en/press/research-news/2011/march/cameras-out-of-the-salt-shaker.html>). Perhaps they should consider highlighting briefly the 'state of the art' in miniature cameras, or detail the specific limitations that metasurface based devices will potentially overcome.

Our response:

The main focus of the manuscript and its main contribution is not to realize the smallest camera, but to demonstrate how monochromatic aberrations of metasurface lenses can be effectively eliminated only by using two cascaded metasurfaces. In addition to its small size and the potential for further miniaturization, the metasurface doublet has several advantages over conventional lens modules that we have now explained more clearly in the discussion section of the manuscript:

"The metasurface-enabled camera we reported here has a flat and thin form factor, small f-number, exhibits nearly diffraction limited performance over a large field of view. From a manufacturing standpoint, the metasurface doublets have several advantages over conventional lens modules. Conventional lens modules are composed of multiple lenses which are separately manufactured and later aligned and assembled together to form the module. On the other hand, the metasurface doublets are batch manufactured with simultaneous fabrication of tens of thousands of doublets on each wafer, and with the metasurfaces aligned to each other using lithographic steps during fabrication. Furthermore, the assembly of the conventional lens modules with the image sensors has to be done in a back-end step, but the metasurface doublet can be monolithically stacked on top of image sensors."

Nevertheless, because the miniature size is one of the attractive features of the metasurface cameras, we also added the following to the Discussion section and compared features of the metasurface doublet and other state of the art miniature lenses:

"Compared to other miniature lenses reported previously [24-27], the metasurface doublet offers significantly smaller f-number and better correction for monochromatic aberrations which lead to brighter images and higher resolution, but they have larger chromatic aberration and narrower bandwidth."

Reviewer #2 comment:

Dielectric metasurfaces for the visible spectral range are promising candidates for novel integrated devices and optical elements. Here, Arabi et al. present a similar concept for realizing a metalens as recently published by Khorasaninejad et al. in Science. However, the work distinguishes slightly as it does not use the Berry phase effect for generating the phase pattern but a propagation effect to obtain the desired phase. A further difference is the material system. The authors used silicon nanostructures in their study which are easier to handle in the fabrication process and would also have advantages for a real commercial application by easier integration with silicon based cameras. I found the idea and demonstration of the entire imaging system with the camera chip very impressive. In my opinion this is the first demonstration of a real application with metasurfaces. Such demonstration of integrated devices was not obtained by the Harvard group.

Our response:

We thank the reviewer for carefully reading the manuscript and for comparing it to a recent work in this field.

Reviewer #2 comment:

The work nicely demonstrates the potential of such dielectric metasurfaces and the realization of a doublet lens system underlines the power of such approaches. Therefore, I will recommend the publication of the manuscript in Nature Communications. I personally believe that the presented approach here will have a greater impact on real imaging systems for particular applications than the recently published work by Khorasaninejad. Overall, the manuscript is well written and the details for the fabrication process are extensively explained. The discussion of the transfer function gives a good inside into the performance of the device.

Our response:

We are glad that the reviewer appreciates the contributions of the current manuscript.

Reviewer #2 comment:

However, I would recommend to move part of the fabrication and the measurement procedure to the supplementary file.

Our response:

The details of fabrication and measurement procedures are currently not part of the main text and are included in the Methods section to comply with the Nature Communications format requirements.

Reviewer #2 comment:

There are only a few comments for the authors which they should take into account for a revision:

On page 6 it is stated that 'The intensity of the image formed by a camera only depends on the NA of its lens.' This statement is in my opinion wrong. The intensity of the image is given by the ratio of the diameter of the entrance pupil to the focal length, which is the inverse f-number.

Our response:

The statement is correct because f-number and NA of a lens corrected for coma and spherical aberrations are related:

$$f\text{-number} = 1/(2NA)$$

[equation (9.3) W. Smith "Modern Optical Engineering," 4th edition, p. 184, McGraw-Hill, 2008.]

We modified the manuscript to clarify this:

"The intensity of the image formed by a camera only depends on the NA of its lens (it is proportional to $1/f\text{-number}^2 = 4NA^2$ [22])."

Reviewer #2 comment:

The distance between the metalens and the CCD chip seems to be important and it was taken care about that in the design as demonstrate in the manuscript. However, I could not find any statement of how the authors did this precise alignment for the measurement? Here it would be helpful to add some information.

Our response:

The metasurface doublet was mounted on a 3-axis translation stage during the measurements. To adjust the distance between the image sensor chip and the doublet, a far object was imaged and distance was adjusted until the image was brought into focus. We added the following explanation to the Methods section:

"The metasurface doublet was mounted on a 3-axis translation stage during the measurements. To set the distance between the image sensor chip and the doublet, a far object was imaged and the distance was adjusted until the image was brought into focus."

Reviewer #2 comment:

It seems like the simulations were performed only with a real part of the refractive index. What is the influence of the imaginary part here? Why can it be neglected?

Our response:

The imaginary part of the amorphous silicon refractive index is smaller than 10^{-4} at 850 nm and is neglected in the simulations. The significantly smaller absorption loss of hydrogenated amorphous silicon compared to crystalline silicon is due to its larger bandgap. We added the following to the Methods section for clarification:

"The imaginary part of the refractive index of amorphous silicon is smaller than 10^{-4} at 850 nm and was ignored in the simulations."

Reviewer #2 comment:

Supplementary Figure 2a shows the laser spectrum. Why is there such a strong modulation in the spectrum that looks like an interference effect?

Our response:

The laser diode used in the characterization is multimode and different peaks observed in the measured spectrum of the laser diode correspond to different Fabry-Perot modes of the laser. We added the following to the caption of Supplementary Figure 2 to clarify this:

“Different peaks observed in the spectrum correspond to different Fabry-Perot modes of the laser cavity.”

Reviewer #3 comment:

Authors report in this paper a compact camera that utilizes a flat metasurface doublet lens to deliver nearly diffraction-limited performance within the field-of-view of 60 degree x 60 degree. The overall dimensions of the camera (including the image sensor) are 1.6 mm x 1.6 mm x 1.7 mm. The metasurface doublet lens consists of one metasurface corrector plate and one metasurface focusing lens. The phase profiles of both metasurface lenses have been optimized to collectively reduce the monochromatic aberrations. As the results, the performance of the metasurface doublet has been significantly improved in comparison with the singlet lens. This is a significant step in developing a high performance flat lens for the purpose of optical imaging, as opposite to the focusing being demonstrated before. Additionally, the metasurface doublet has been conveniently fabricated on the both side of 1mm thick quartz substrate with the alignment accuracy of 2 μm . It eliminates the post-fabrication alignment procedure and thus, makes it possible for potential vertical integration using the well-established micro-fabrication capabilities.

Our response:

We thank the reviewer for summarizing the manuscript, and we are glad that realize the impact of the manuscript on imaging using metasurfaces.

Reviewer #3 comment:

Just for the curiosity, can author explain of the whether the functions of the two metasurface lenses can be combined into one metasurface lens with aspherical phase profile?

Our response:

The singlet metasurface lens that we used for comparison is aspheric and the only possible design with no spherical aberration. As we showed in the manuscript a metasurface lens corrected for spherical aberration (i.e. the aspheric singlet) has significant coma, so it is not possible to make a singlet which is corrected for both spherical and coma aberrations.

Reviewer #3 comment:

Furthermore, the title of “An optical metasurface planar camera” is not very accurate. The demonstrated camera has the shape close to a cubic so it is hard to define it as a “flat camera”. The lens being used is the flat one though.

Our response:

The term “planar” in the title refers to the planar metasurface lenses made using planar fabrication technology, and does not mean that the camera is infinitesimally thin. To eliminate the confusion and to make the title more descriptive, we changed the title to:

“Miniature optical planar camera based on a wide-angle metasurface doublet corrected for monochromatic aberrations”

Reviewer #3 comment:

Overall, the manuscript is well written and the reported work is of the great interest to the readers. I would recommend the paper to be accepted for publication with minor revision.

Our response:

We thank the reviewer for providing constructive feedback and recommending the manuscript for publication.

Reviewer #4 comment:

The authors describe in their paper a planar single-layer and double-layer lens based on optical metasurfaces. Furthermore they combine their proposed metasurface doublet lens with a commercial CMOS image sensor.

The used approach for producing the metasurface lenses is not restricted to laboratory prototypes. A high-volume fabrication with thin-film production line is possible, which might yield to a high economic impact of the proposed lenses.

Additionally the authors combine their proposed lens with a commercial CMOS image sensor. Compared to the state-of-the-art they replace a "classical" lens with their proposed one.

Our response:

We thank the reviewer for carefully reading the manuscript and summarizing its results.

Reviewer #4 comment:

However the authors should more clearly explain the advantages of their lens for the camera module. From a production point of view for the camera module I do not see a significant advantage. The

assembly of the lens together with the CMOS sensor still needs to be done in the backend with a very similar process.

Our response:

From a production point of view, the metasurface doublet lenses have several advantages over conventional lens modules with similar degree of corrections for monochromatic aberrations. A conventional lens module is made of multiple lenses which are separately manufactured and later aligned and assembled together. The metasurface doublets are batch manufactured with the potential for simultaneous manufacturing of tens of thousands of doublet lenses on the same wafer, and the two lenses of the doublet are aligned using a single lithographic step during fabrication. Furthermore, the assembly of the conventional lens modules with the image sensor has to be done in a back-end step, but the metasurface doublet has the potential for monolithic integration with the image sensor. We added the following explanation to the Discussion section of the manuscript to further emphasize these advantages:

“From a manufacturing standpoint, the metasurface doublets have several advantages over conventional lens modules. Conventional lens modules are composed of multiple lenses which are separately manufactured and later aligned and assembled together to form the module. On the other hand, the metasurface doublets are batch manufactured with simultaneous fabrication of tens of thousands of doublets on each wafer, and with the metasurfaces aligned to each other using lithographic steps during fabrication. Furthermore, the assembly of the conventional lens modules with the image sensors has to be done in a back-end step, but the metasurface doublet can be monolithically stacked on top of image sensors.”

Reviewer #4 comment:

A topic not mentioned in the paper is the influence of the alignment of the nano-posts of the lens and the bayer pattern of the CMOS sensor.

Our response:

There is no need for aligning the Bayer filter pattern on the image sensor and the nano-posts. Generally, the lens module of a camera does not need to be aligned with the Bayer pattern on the image sensor. This applies to the metasurface doublets as well, because the nano-posts collectively form lenses which function similar to conventional glass lenses. Furthermore, the metasurface doublet is designed for a single color and should be used with a monochrome image sensor.

Reviewer #4 comment:

Some further remarks:

- Fig. 2b: what's the reason for the low transmission values

Our response:

The periodic array of nano-posts exhibits distributed resonances for the diameter values corresponding to low transmission. We added the following to the caption of Fig. 2 to clarify this point:

“The diameters with low transmission values, which are highlighted by two gray rectangles, correspond to distributed resonances of the periodic array of nano-posts, and are excluded from the designs.”

Reviewer #4 comment:

- Non consistent wording for the substrate of the lens: "fused silica" vs. "glass" might be a bit confusing

Our response:

To eliminate any potential confusion, we replaced all the instances of “glass” with “fused silica” in the revised manuscript.

Reviewer #4 comment:

- p. 7: "high-throughput nano-fabrication techniques" is in my opinion a bit misleading. The fabrication of the lens together with the CMOS sensor in the frontend is in my opinion not feasible. Instead both have to be produced separately in different frontend processes and then combined in the backend

Our response:

The “high-throughput nano-fabrication techniques” applies to the fabrication of the metasurface doublet and not to its integration with the image sensor. We believe that the metasurface doublet can be bonded to the cover glass wafer and serve as both the cover glass of image sensors and the imaging optics. Nevertheless, in response to the previous comment of the reviewer on advantages of the doublet over conventional lens module, we revised the manuscript and clarified this point:

“From a manufacturing standpoint, the metasurface doublets have several advantages over conventional lens modules. Conventional lens modules are composed of multiple lenses which are separately manufactured and later aligned and assembled together to form the module. On the other hand, the metasurface doublets are batch manufactured with simultaneous fabrication of tens of thousands of doublets on each wafer, and with the metasurfaces aligned to each other using lithographic steps during fabrication. Furthermore, the assembly of the conventional lens modules with the image sensors has to be done in a back-end step, but the metasurface doublet can be monolithically stacked on top of image sensors.”

Reviewer #4 comment:

All in all the paper describes a novel and interesting approach for planar lenses which the possibility for a high impact. In addition the authors describe a planar camera module with their planar lenses. This approach is also interesting, but the advantages of the proposed solution are not completely clear for me.

Our response:

We are glad to see that the reviewer has a positive opinion about the work and realizes its high impact. The main advantages of the are the high imaging quality with small and flat form factor, high scalability

of the fabrication process allowing for batch fabrication of a large number of camera lenses on a same wafer, and elimination of post-fabrication alignment and assembly steps required for fabrication of camera lenses. The advantages of the metasurface doublet over conventional design, and the broader impact of the vertical integration approach introduced in the manuscript are now explained more clearly in the last paragraph of the revised manuscript:

“The metasurface-enabled camera we reported here has a flat and thin form factor, small f-number, exhibits nearly diffraction limited performance over a large field of view. From a manufacturing standpoint, the metasurface doublets have several advantages over conventional lens modules. Conventional lens modules are composed of multiple lenses which are separately manufactured and later aligned and assembled together to form the module. On the other hand, the metasurface doublets are batch manufactured with simultaneous fabrication of tens of thousands of doublets on each wafer, and with the metasurfaces aligned to each other using lithographic steps during fabrication. Furthermore, the assembly of the conventional lens modules with the image sensors has to be done in a back-end step, but the metasurface doublet can be monolithically stacked on top of image sensors. More generally, this work demonstrates a novel vertical on-chip integration architecture for designing and manufacturing optical systems, which is enabled through high performance metasurfaces. This architecture will enable low-cost realization of conventional optical systems (e.g. spectrometers, 3D scanners, projectors, microscopes, etc.), and systems with novel functionalities in a thin and planar form factor with immediate applications in medical imaging and diagnostics, surveillance, and consumer electronics.”

REVIEWERS' COMMENTS:

Reviewer #1 (Remarks to the Author):

The authors have made satisfactory amendments to the manuscript in response to my previous comments. I also note that great effort has been made to appease the other reviewer comments. Overall the manuscript reads well, has clarity, and communicates the work of the authors. In my opinion this manuscript is suitable for publication in Nature Communications.

Reviewer #2 (Remarks to the Author):

In the revised manuscript the authors carefully addressed the raised questions and concerns. The manuscript contains now all information and can be accepted for publication. I also find the new title more appealing than the old one. Therefore, I support the change in the title of the paper.

Reviewer #4 (Remarks to the Author):

The authors have addressed all of my comments and I recommend to accept the paper.

Reviewer #1 (Remarks to the Author):

The authors have made satisfactory amendments to the manuscript in response to my previous comments. I also note that great effort has been made to appease the other reviewer comments. Overall the manuscript reads well, has clarity, and communicates the work of the authors. In my opinion this manuscript is suitable for publication in Nature Communications.

Reviewer #2 (Remarks to the Author):

In the revised manuscript the authors carefully addressed the raised questions and concerns. The manuscript contains now all information and can be accepted for publication. I also find the new title more appealing than the old one. Therefore, I support the change in the title of the paper.

Reviewer #4 (Remarks to the Author):

The authors have addressed all of my comments and I recommend to accept the paper.

We thank the reviewers for considering our response and recommending the manuscript for publication.